Host specificity of gastrointestinal parasites in free-ranging sloths from Costa Rica

Vanderhoeven Ezequiel A. 1 2 3
Florida Madeleine 1 3
Cliffe Rebecca N. 4
Guzmán José 4
Notarnicola Juliana 2 5
http://orcid.org/0000-0002-8488-0580 Kartzinel Tyler R. 1 3 tyler_kartzinel@brown.edu
1 Department of Ecology, Evolution, and Organismal Biology, Brown University , Providence, Rhode Island , United States
2 Instituto de Biología Subtropical, Consejo Nacional de Investigaciones Científicas y Técnicas, Universidad Nacional de Misiones , Puerto Iguazu, Misiones , Argentina
3 Institute at Brown for Environment and Society, Brown University , Providence, Rhode Island , United States
4 Sloth Conservation Foundation , Puerto Viejo de Talamanca , Costa Rica
5 Laboratorio de Diagnóstico Vegetal y Animal, Facultad de Ciencias Forestales, Universidad Nacional de Misiones , Eldorado, Misiones , Argentina
Manjarrez Javier
Electronic publication date: 2025 May 8
Publication date: 2025
Volume: 13
Electronic Location ID: e19408
Received 2025 Jan 9; Accepted 2025 Apr 10
Copyright: © 2025 Vanderhoeven et al.
Copyright year: 2025
Copyright holder: Vanderhoeven et al.
License: This is an open access article distributed under the terms of the Creative Commons Attribution License, which permits unrestricted use, distribution, reproduction and adaptation in any medium and for any purpose provided that it is properly attributed. For attribution, the original author(s), title, publication source (PeerJ) and either DOI or URL of the article must be cited.
License URL: https://creativecommons.org/licenses/by/4.0/

Keywords: Central America, Cestoda, Three-toed sloth, Two-toed sloth, Nematoda, Parasite communities

Funding: Tropical Studies Early Career Rexford Daubenmire Fellowships 514/564 1448 Caleel Undergraduate Fellowship Brown Brown University’s Office of the Vice President for Research Funding was provided by the Organization for Tropical Studies Early Career Rexford Daubenmire Fellowships, (No. 514/564 1448) The 2023 Caleel Undergraduate Fellowship Brown and the 2024 Brown University’s Office of the Vice President for Research. The funders had no role in study design, data collection and analysis, decision to publish, or preparation of the manuscript.

==============================
The diversity and host specificity of gastrointestinal parasites infecting free-ranging sloths is poorly known. We compared gastrointestinal parasites of two sloth species from Costa Rica—three-fingered sloths (Bradypus variegatus) and two-fingered sloths (Choloepus hoffmanni)—for the first time in both a primary forest and an urban habitat. We asked whether host-parasite interactions were predominantly structured by host identity, the habitats in which hosts occurred, or both. Coproparasitology revealed protozoa and nematode eggs from both host species, but cestode eggs were recorded only in C. hoffmanni. We found eight parasitic morphotypes in 38 samples, which matches the total number of these parasites described in sloths over the past 100 years. We found no significant difference in overall parasite richness between sloth species or habitats, but the parasite richness of C. hoffmanni was 2-fold greater in the primary forest vs. urban habitat. As no parasite sharing was observed between sloth species, we found strong and significant differences in parasite composition between host species regardless of habitat. In B. variegatus, we observed eggs of four nematode taxa (Spirocercidae, Subuluroidea, Spirurida, Ascaridida) and cysts of Eimeriidae (Apicomplexa). By contrast, in C. hoffmanni, we observed cestodes (Anoplocephalidae), a different nematode from the family Spirocercidae, and also different cysts of Eimeriidae (Apicomplexa). Many rare taxa were recorded only in samples from the primary forest, and these did not match any sloth parasites that had been previously described in the literature, suggesting that at least some could be undescribed species. Together, these results highlight the paucity of comparative parasitology involving tropical wildlife, the importance of characterizing host-parasite transmission networks, and the potential relevance of intermediate hosts that may be relevant to sloth health.

Introduction

Parasites are defined as organisms that live in or on other organisms and obtain nutrients from their hosts, living all or part of their lives at the host’s expense—regardless of whether their tendency to cause disease or otherwise harm their host is consistent or severe (Méthot & Alizon, 2014; Rózsa & Garay, 2023). While organisms with parasitic life cycles can be commensal or, in rare cases, conditionally beneficial to their hosts (Fellous & Salvaudon, 2009), research in disease ecology has repeatedly shown that host susceptibility to harmful parasites can be strongly influenced by the context in which each host population occurs, and that exposure to parasites can profoundly affect host health and survival (Preston & Johnson, 2010; Irvine, 2006; Wood et al., 2023). Theoretically, the parasite assemblage associated with co-occurring host species from divergent mammalian lineages could be structured primarily by host phylogeny and any associated functional differences in anatomy, immunity, and/or behavioral ecology (Morand, 2015). Yet, although the study of gastrointestinal (GI) parasites is common in domestic animals, for which taxonomic keys facilitate identification, there is a scarcity of even coprological research into the GI parasites of wildlife mammals (Thienpont, Rochette & Vanparijs, 1979; de Almeida Curi et al., 2010; Kaufmann, 2013; Panayotova-Pencheva, 2013; Solórzano-García & de León, 2017; Agostini et al., 2018; Červená et al., 2024; Rondón et al., 2025).

Two- and three-toed sloths are distantly related members of the Pilosa (Xenarthra) order. Each sloth lineage evolved anatomical and behavioral characteristics associated with their slow, herbivorous, arboreal lifestyles from a terrestrial common ancestor via convergent evolution (Chiarello, 1998; Gaudin, 2003; Toledo et al., 2015; Hayssen, 2010, 2011; Urbani & Bosque, 2007). Fossil sloths were diverse and geographically widespread, extending in the southernmost Chile, Argentine Patagonia, and possibly Antarctica in the south and to the south of Alaska in the north (Toledo et al., 2015). Two- and three-toed sloth lineages diverged ~30 million years ago, and since the megafaunal extinction of the terminal Pleistocene they have maintained broadly overlapping ranges across Central and South America (Toledo et al., 2015). Sloths are keystone herbivores of the Neotropical forests and today their populations occur in a diversity of habitats including forests, agroecosystems, and urban environments where they may often come into close contact with humans or domestic animals—creating opportunities for GI parasite transmission (Brandão et al., 2019; Hayssen, 2010, 2011; Smith & Ruple, 2021; Superina et al., 2010; Superina & Loughry, 2015). Yet most prior studies of sloth GI parasites have involved captive animals, with only a few reports of necropsies involving wild animals (Diniz & Oliveira, 1999; Sibaja-Morales et al., 2009; Dünner & Pastor, 2017; Michel et al., 2017; Araujo et al., 2021). Studies of wild sloth parasites have tended instead to focus on their potential role as reservoirs for Leishmania braziliensis and other blood parasites such as Trypanosoma spp. and microfilariae that pose risks to humans (Gilmore, Da Costa & Duarte, 2001; Herrer & Christensen, 1980; Lainson et al., 1989; Muñoz-García et al., 2019; Sant’Ana et al., 2020). Gaining knowledge of GI parasites circulating in wild sloth populations is thus a research priority at the intersection of wildlife ecology, conservation, and public health (Carlson et al., 2020; Pedersen & Fenton, 2007).

In Costa Rica, two sloth species are distributed across a range of environmental conditions: Hoffmann’s two-toed sloths (Choloepus hoffmanni Peters, 1858) and the brown-throated three-toed sloths (Bradypus variegatus Schinz, 1825; Santos et al., 2019; Superina et al., 2010). Given their broadly overlapping ecologies and their many convergent anatomical and behavioral characteristics—especially in their unique digestive morphophysiologies—wild populations of these co-occurring species in different habitats provide unique opportunities to evaluate the extent to which GI parasite assemblages are structured primarily by host identity or their shared ecological conditions. Therefore, we tested the hypotheses that the diversity and composition of GI parasites communities infecting these two species of wild sloths are determined primarily by (1) host species identity, (2) local habitat type, or (3) that they are modulated by both. Results will address a critical research need concerning our basic understanding of host-parasite interactions in tropical wildlife, the ability to effectively monitor wildlife diseases and potential zoonoses, and the success of numerous conservation activities in the region.

Portions of this text were previously published in a preprint (Vanderhoeven et al., 2024).

Materials and Methods

Study sites

We collected samples during March–July 2023 across the Caribbean coast of Costa Rica (Fig. 1). Principal study sites included La Selva Biological Station and the city of Puerto Viejo de Talamanca. La Selva is a field station operated by the Organization for Tropical Studies (OTS) in Sarapiquí of northeastern Costa Rica, comprising 1,536 hectares of protected lowland tropical rainforest (10°25.32′N, 84°00.9′W; elevation 37–150 mASL) adjoining the 47,000-hectare Braulio Carrillo National Park (Fig. 1; Matlock & Hartshorn, 1999). Puerto Viejo de Talamanca (9.65°N, 82.75°W, elevation 0–4 mASL) is a small, but densely urbanized coastal town in southeastern Costa Rica, near both Cahuita and Manzanillo National Parks (Fig. 1; Lindshield, 2016; Emard & Nelson, 2021; Fan & Lindshield, 2022).

Figure 1 Map of Costa Rica.

Insets showing the two main sites of sample collection, including La Selva Biological Station (Primary Forest) to the northwest and the city of Puerto Viejo de Talamanca (Urban habitat) to the southeast. The green shading shows areas of tree cover and the dotted areas in the northern inset show Braulio Carillo National Park, which is contiguous with La Selva Biological Station. In contrast, in Puerto Viejo the grey areas evidence the degree of urbanization and the loss of forest.

Sample collection

We obtained samples from sloth feces using two sampling methods. First, we conducted active searches on foot through trails and roads for signs of sloth feces. The feces of both species of sloths have a distinctive shape and size that allow us to identify host species (Fig. S1, Appendix S1). Most samples were collected by searching the bases of trees in both primary and urban forest sites that wild sloths used as latrines. We selected fecal pellets from the center of the bolus, avoiding those in contact with the surface or soil, and used qualitative criteria such as the absence of fungi, moisture, and shininess. We also obtained a subset of urban sloth samples from veterinarians at wildlife hospitals, who provided the first fecal sample obtained from animals that had been admitted for care. These samples provided by third-party veterinarians allowed us to enhance our sample size for urban sloths without the risk of confounding variables associated with nutritional or medical changes while under care. All samples were placed in 125-mL plastic containers with 50 mL of 5% formalin for parasitological analyses and stored at room temperature until analysis. This research was conducted with permits provided by the National System of Conservation Areas of Costa Rica (SINAC-ACC-PI-LC-052-2023).

Parasitology techniques

To detect a wide variety of parasitic structures including helminth eggs, larvae, and protozoan cysts, we used a set of techniques involving concentration, sedimentation, and flotation. First, we used the modified Telemann method, which is a sedimentation and concentration technique that allows visualization of parasitic structures such as heavy and light eggs, cysts, and larvae, especially in samples that have high concentrations of neutral fats and free fatty acids (Thienpont, Rochette & Vanparijs, 1979). The procedure consisted of homogenizing 1–2 g of fecal material preserved in formalin with water (5–6 pellets from B. variegatus; 1–3 pellets from C. hoffmanni). This was filtered through a strainer over a funnel, and 10 mL of filtrate was collected in a 15-mL conical tube to which 2 mL of sulfuric/ethyl ether was added and shaken (stopping to vent halfway through and after shaking). The resulting mixture was centrifuged at 1,500 rpm for 5 min to pellet the parasitic elements before removing the supernatant. The pellet was placed on a slide for observation using a Pasteur pipette and stained with a drop of 1% Lugol’s solution for visualization. Second, a modified Sheather method involved the use of a supersaturated sugar solution view light helminth eggs and oocysts of Cryptosporidium spp. and Giardia spp. (Sheater, 1923). It began by homogenizing 1–2 g of fecal material in water, followed by filtration through a strainer placed over a funnel from which 10 mL of filtrate was collected in a 15 mL conical tube. The sample was centrifuged at 2,500 rpm for 2 min, the supernatant discarded, and the tube filled with Benbrok solution (density 1.300) to 1 cm from the rim. The tube was shaken until the sediment dissolved and was then centrifuged at 1,000 rpm for 2 min. More Benbrok solution was added until a dome held in place by surface tension formed above the lip of the tube. A coverslip placed on top of the tube was left to incubate for 10 min before observation on a microscope at 10× and 40× magnification. We used an AmScope B400 Series microscope and took pictures with an AmScope MU1400 CMOS C-Mount Microscope Camera with Reduction Lens. Parasite structures were measured using ImageJ and eggs were identified to the finest taxonomic resolution possible according to relevant taxonomic keys and specific literature on helminths from sloths (Kaufmann, 2013; Vicente et al., 1997; Anderson, 2000; Khalil, Jones & Bray, 1994).

Statistical analysis

First, we tested for statistically significant differences in mean parasite richness according to sloth species, habitat, and the sloth species × habitat interaction using ANOVA. We based this richness analysis on all 38 samples, including 13 that tested negative for parasites and therefore had richness scores of zero. Then, after excluding the 13 samples in which no parasites were identified, we quantified pairwise similarity among the 25 positive sloth samples using the Jaccard Dissimilarity Index (0 = complete overlap; 1 = mutually exclusive). We tested for significant differences in parasite composition according to sloth species, habitat, and the sloth species × habitat interaction using permutational MANOVA (PERMANOVA) with 999 permutations as implemented in adonis2 in vegan (Jaccard, 1912; Oksanen et al., 2024). This PERMANOVA approach partitions sums of squares of the multivariate dataset based on the Jaccard dissimilarities and is a robust alternative to ordination for quantifying how variation can be attributed to grouping variables (Anderson, 2001). It is sensitive to both differences in the within-group variation (i.e., dispersion) as well as the mean location of groups in multivariate space (i.e., centroids), both of which can contribute to significant differences between groups (Warton, Wright & Wang, 2012). All statistical analyses were performed in R (R Development Core Team, 2023).

Results

Parasite identification and prevalence

We obtained 22 fecal samples from B. variegatus and 16 from C. hoffmanni, including all samples provided by third-party veterinarians (Appendix S1, Table S2). Of the 22 B. variegatus samples (N = 12 primary forest; 10 urban), a total of 64% tested positive for parasites (58.8% primary forest; 66.6% urban; Table S2). Of the 16 C. hoffmanni samples (N = 5 primary forest; 11 urban), we found a total of 69.0% tested positive (80% primary forest; 64% urban; Table S2). We classified parasite structures into eight morphotypes based on a thorough review of the relevant literature (Table 1; Table S1; Appendix S1). In B. variegatus, we identified four distinct types of Nematoda eggs and protozoa of the family Eimeriidae (Apicomplexa; Table 1; Fig. 2). In C. hoffmanni, we observed one morphotype of Nematoda, one morphotype of Anoplocephalidae (Cestoda), and cysts of Eimeriidae (Apicomplexa; Fig. 3). In both host species, Eimeriidae cysts showed the highest prevalence (Table 1).

Table 1 Gastrointestinal parasites of sloths.

Eggs characterized as GI parasites were grouped into morphotypes and identified to the finest taxonomic level possible within three phyla and five families. Representative photos are shown in Figs. 2 and 3, along with mean length × width measurements (with parentheses indicating the number of eggs measured), shapes, general descriptions, habitat, and prevalence.

Host	Egg morphotype	Phylum	Size (μm)	Shape	Description	Prevalence (%)	
Primary Forest	Urban	Total	
Bradypus variegatus
NPrimary forest = 12
NUrban = 10
NTotal = 22	Spirocercidae egg morphotype 1
(Fig. 2A)	Nematoda	38.8 × 18.6 (1)	Oval	Thin shell, larval eggs	0.0	10.0	4.2	
Spirurida egg morphotype (Fig. 2B)	Nematoda	50.0 × 23.7 (12)	Oval	Thick, smooth shell, larval eggs	16.7	40.0	27.3	
Subuluroidea egg morphotype
(Fig. 2C)	Nematoda	40.5 × 24.1 (2)	Oval	Smooth thin shell, no larval development	8.3	0.0	4.2	
Ascaridida egg morphotype
(Fig. 2D)	Nematoda	77.9 × 63.0 (2)	Rounded	Thick shell, larva visible inside,	16.7	0.0	9.1	
Coccidia cyst morphotype 1
(Fig. 2E)	Apicomplexa	25.4 × 23.4 (10)	Round	Thin shell	25.0	40.0	31.9	
Choloepus hoffmanni
NPrimary forest = 5
NUrban = 11
NTotal = 16	Anoplocephalidae morphotype 1
(Fig. 3A)	Platyhelminthes	47.8 × 36.1 (12)	Triangular to oval	Thick wall, visible pyriform apparatus	60.0	36.4	43.8	
Spirocercidae egg morphotype 2
(Fig. 3B)	Nematoda	49.0 × 23.6 (2)	Oval	Thin double shell	20.0	0.0	6.3	
Coccidia cyst morphotype 2
(Fig. 3C)	Apicomplexa	22.3 × 18.7 (16)	Rounded	Thin shell	80.0	36.6	56.3	

Figure 2 Images of parasite groups identified in B. variegatus.

The figure shows morphology and size. (A) Spirocercidae egg morphotype 1, (B) Spirurida egg morphotype, (C) Subuluridae egg morphotype, (D) Ascaridida egg, (E) Coccidia cyst morphotype 1.

Figure 3 Images of parasite groups identified in C. hoffmanni.

The figure shows morphology and size. (A) Anoplocephalidae morphotype, (B) Spirocercidae egg morphotype 2, (C) Coccidia cyst morphotype 2.

Statistical results

We found no significant difference in mean parasite richness between sloth species (ANOVA: F1,34 = 1.3, P = 0.263) or habitat types (F1,34 = 0.4, P = 0.544), but there was a marginally significant sloth species × habitat type interaction (F1,34 = 3.6, P = 0.066). This interaction reflected the elevated parasite richness in C. hoffmanni samples from primary forest habitats (Fig. 4). This effect of habitat on C. hoffmanni was evidently strong, albeit statistically underpowered, as the mean parasite richness was ~2-fold greater in primary forest habitats compared to urban habitats (Fig. 4). In contrast to patterns of parasite richness across sloth species, the composition of parasites observed in B. variegatus differed profoundly from C. hoffmanni. There was a strong and statistically significant difference in parasite composition between sloth species (PERMANOVA: pseudo-F1,21 = 13.0, R2 = 0.37, P ≤ 0.001), but not between habitat (pseudo-F1,21 = 0.7, R2 = 0.02, P = 0.587), and there was no significant sloth species × habitat interaction (pseudo-F1,21 = 0.6, R2 = 0.02, P = 0.716). As we observed no shared parasites between sloth species, mean Jaccard dissimilarity between these host species was 1.00, far exceeding the mean differences among samples within species overall (B. variegatus = 0.71, C. hoffmanni = 0.46) and between habitats (urban habitat vs. primary forest for B. variegatus = 0.70, C. hoffmanni = 0.43; Table 1).

Figure 4 Parasite richness barchart.

Mean parasite richness across sloth populations in each species-habitat combination. Mean parasite richness across sloth species and habitat types. No significant differences were observed between species (F1,34 = 1.3, P = 0.263) or habitats (F1,34 = 0.4, P = 0.544), but a marginally significant interaction (F1,34 = 3.6, P = 0.066) indicates higher richness in C. hoffmanni from undisturbed primary forest. Error bars represent standard error.

Discussion

We asked whether sloth-parasite interactions are structured primarily by host identity or the local environment. Our comparisons of two co-occurring sloth species from regions characterized by different levels of human impacts revealed five parasitic morphotypes from B. variegatus and three from C. hoffmanni. The parasite communities were mutually exclusive to sloth species, even between populations co-occurring in the same sites, suggesting a high degree of host specificity in these host-parasite interaction networks. Furthermore, many parasitic morphotypes were shared between conspecific populations that were geographically and ecologically separated across primary forest and urban habitats. Our results demonstrate a strong role for host species identity in structuring the composition of parasite assemblages in sloths.

Our study is the first to report GI parasites from primary-forest sloths, reflecting the long-standing difficulty of obtaining samples from sloth populations in structurally complex habitats compared to more human-dominated environments such as urban green spaces and agricultural areas. Although our sample set from the primary forest at La Selva was relatively small (N = 17), it supports several important conclusions (Table 1; Fig. 4). First, there was no obviously consistent effect of habitat on the composition of sloth parasites across both species. This is in part because we encountered so many undescribed helminth taxa in sloths from primary forest, especially for B. variagatus, that we had a limited ability to discern whether these taxa comprise a community of parasites that is characteristic of primary forests or if they were simply rare and potentially opportunistic parasites of the sloths we encountered. Second, the mean parasite richness of C. hoffmanni was strongly, albeit marginally significantly, greater in primary forest than urban habitats—a trend driven largely by the elevated prevalence of cestodes in primary forest sloths. Third, although protozoa are commonly associated with elevated stress levels (Fayer, 1980), and although it would be reasonable to hypothesize that sloths experience elevated stress in human-dominated environments, we did not observe elevated protozoa of the family Eimeriidae (Apicomplexa) infection rates in urban sloths (41% of 17 primary forest samples vs. 43% of 21 urban samples). These outcomes could reflect differences in the abundance and diversity of intermediate hosts supporting parasite lifecycles, environmental variation in sloth diet composition and nutrition, and/or differences in the timing or extent to which GI infections spread through geographically discrete populations. For example, the transmissibility of each parasite can be modulated by both external environmental factors (e.g., contact between hosts in continuous vs. fragmented habitats, seasonal variations) or internal variability in host susceptibility to infection (e.g., disparate levels of nutrition or environmental stress; Pool et al., 2016; Silva et al., 2013; Werner & Nunn, 2020).

Even with a conservative approach to differentiating parasitic morphotypes, our study revealed greater richness of sloth GI richness from these 38 samples than all prior publications combined (Sibaja-Morales et al., 2009; Vicente et al., 1997). Our results raise questions about how sloths ingest such a diversity of parasites, especially since the predominant parasitic taxa infecting both host species are expected to rely on intermediate hosts to complete their lifecycles. We can consider at least three non-mutually exclusive hypotheses: (i) although documented almost exclusively in captivity, sloths are capable of omnivory, and they may occasionally scavenge on tissues of animals serving as intermediate hosts, (ii) sloths incidentally ingest arthropods or their frass while foraging on contaminated vegetation (e.g., floral mites), or (iii) sloth-associated arthropods such as moths and beetles act as carriers that transport infective materials from sloth stool on the forest floor to susceptible hosts in the canopy (Denegri et al., 1998; Madrigal, 2020; Wickström, 2004). We cannot differentiate between hypotheses ii or iii, but several lines of evidence suggest hypothesis i is insufficient to account for the prevalence of parasites that require intermediate hosts infecting both species. First, although it is known that C. hoffmanni can engage in omnivory because they accept eggs in captivity (Reyes-Amaya et al., 2015), and although prior studies of sloth diets have focused almost exclusively on herbivory (Sánchez-Chavez, 2021; Vaughan et al., 2007), and there is no evidence of omnivory in B. variegatus that could explain the diversity and prevalence of Spirocercidae infecting both species. Second, arthropods represent the most likely intermediate hosts of these parasitic taxa: common intermediate hosts of Spiruroidea include beetles and other arthropods (Bain, Mutafchiev & Junker, 2013; Chabaud & Bain, 1994), while oribatid mites or collembolans may often serve as intermediate hosts for Anoplocephalidae cestodes (Wickström, 2004). Sloth-associated beetles, mites, and moths live in sloth fur, for example, and some of these arthropods oviposit in sloth feces, creating ample opportunities to transfer infectious material to sloths for ingestion. Additional work will be needed to differentiate between hypotheses ii and iii, especially as both direct (e.g., arthopod carriers) and indirect (e.g., food-borne) routes of fecal-oral transmission are plausible arthropods to serve as intermediate hosts of these GI parasites could help elucidate transmission pathways.

Coproparasitology works well when the focus of a study is comparative because it allows for internal consistency of taxonomic identifications. This method enabled us to compare the parasite taxa of two co-occurring sloth species across an environmental gradient, providing the first evidence for important patterns in the sloth-parasite interaction network and highlighting several productive avenues for future research. By contrast, it is notoriously difficult to compare across studies, and some imprecision in identifications is inevitable (Akrim et al., 2018: Rojas et al., 2024). For each of the eight morphotypes documented here, we reviewed what was previously documented in sloths and identified 11 relevant articles reporting both egg structures and adult parasites—these included articles dating to 1928 and an unpublished student thesis (Appendix S1). Five of the eight taxa that we observed were similar to ones presented in these previous reports, but we could find no evidence of the other three being previously reported in association with sloths (Appendix S1). This review of the prior literature highlights the strengths and weaknesses of coproparasitological approaches, which yield only coarse taxonomic identifications but can be successfully implemented in a diversity of wild animals (e.g., Kooriyama et al., 2012; Parr, Fedigan & Kutz, 2013; Rondón et al., 2017; Agostini et al., 2018; Solórzano-García & de León, 2017; Hou et al., 2020; Hewavithana, Wijesinghe & Udagama, 2022; Bellusci et al., 2024). Emerging molecular techniques such as DNA barcoding present new opportunities to simultaneously improve taxonomic precision, enhance sample sizes, forensically link the primary and intermediate hosts of parasites across life stages, and facilitate comparisons across sites and species (Titcomb et al., 2022).

New research priorities to address knowledge gaps concerning the parasites of wild sloths present several exciting possibilities. First, given the high diversity of novel parasites we observed, and given the difficulty of precisely identifying parasites when observing their eggs, necropsies to collect and properly identify adult parasites would enable more comprehensive efforts to compare the parasites of free-ranging sloths. Studies of Bradypus spp. in Brazil, for example, have characterized five adult gastrointestinal nematode species (Michel et al., 2017; Santos & Werneck, 2013; Werneck et al., 2008). A lack of adult nematode specimens that can be positively identified leaves us uncertain if the parasites infecting Bradypus in Costa Rica are the same as have been documented in Brazil. Second, evaluations of how GI parasites affect sloth health would require veterinary assessment. The parasites that we identified likely differ in the level and consistency harm they cause to their hosts—if any—which could vary as a function of both parasite diversity and load. Indeed, the mutualism-parasitism continuum holds that host-parasite interactions can vary from detrimental to commensal or beneficial depending on environmental context (Bronstein, 1994). Combinations of methods including fecal DNA analysis and egg counts could help better characterize the abundance, diversity, and potential health risks posed by parasites throughout the range of sloths (Chan et al., 2022; Titcomb et al., 2022). Finally, whereas our study highlighted the diversity of sloth parasites that may be discovered in primary forests, our comparisons were based on just two geographically discrete habitats and further comparative studies could help clarify or reinforce the differences in parasite diversity that we observed here.

Conclusion

We provide evidence that host identity is the main driver to parasite communities in sloths. The absence of significant habitat influences on parasite composition and the unexpectedly high richness of parasites in primary forests point to ecological complexities that merit further studies. Coprological analyses provide valuable initial insights into parasite communities, but research that integrates necropsy-based parasite identification, and broader habitat comparisons are essential to address the taxonomic uncertainties and ecological questions raised by our findings. Lastly, sloths are among the species most frequently admitted to veterinary clinics for treatment and release following encounters with domestic wildlife, electrocution, or poor health. Although both species are listed as Least Concern (LC) on the IUCN red list (Moraes-Barros et al., 2022; Plese et al., 2022), some suggest that C. hoffmanni would be reclassified as threatened if more data were available (Sánchez-Chavez, 2021). Considering these many newfound sloth-parasite interactions and the unknown health effects they may have, caution to minimize the risk of transmitting harmful parasites between populations and species may be warranted when undertaking ecological rewilding programs, translocations, or ex situ care of wild sloths.

Supplemental Information

Supplemental Information 1 Size and shape differences between fecal pellets of each sloth species.

The scale is shown in millimeters.

Supplemental Information 2 Gastrointestinal parasite species that have been identified based on adult specimens in association with sloths in prior publications.

Supplemental Information 3 Raw data for each fecal sample included in the analysis.

A unique identifier for each fecal sample, with information on sloth species, habitat, collection date, the presence of any parasite (positive/negative), the parasite morphotype(s) detected, and the total parasite richness in cases of coinfection.

Supplemental Information 4 Studies of gastrointestinal parasites from sloths are limited, but nevertheless prior publications provide a basis for interpretation of our results.

We thank Enrique Castro, Orlando Vargas, Danilo Brenes, Dr. Dennise Ortiz, Uxia Rico Gomez, Dr. Jorge Mercardo, Toucan Rescue Ranch, Kids Saving the Rainforest, the Sloth Sanctuary, and the Alturas Wildlife Sanctuary for facilitating our research. ChatGPT (OpenAI) was used for language editing and grammar checking.

Additional Information and Declarations

Competing Interests

The authors declare that they have no competing interests.

Author Contributions

Ezequiel A. Vanderhoeven conceived and designed the experiments, performed the experiments, analyzed the data, prepared figures and/or tables, authored or reviewed drafts of the article, and approved the final draft.

Madeleine Florida performed the experiments, analyzed the data, prepared figures and/or tables, and approved the final draft.

Rebecca N. Cliffe conceived and designed the experiments, authored or reviewed drafts of the article, and approved the final draft.

José Guzmán conceived and designed the experiments, performed the experiments, authored or reviewed drafts of the article, and approved the final draft.

Juliana Notarnicola analyzed the data, authored or reviewed drafts of the article, and approved the final draft.

Tyler R. Kartzinel conceived and designed the experiments, analyzed the data, prepared figures and/or tables, authored or reviewed drafts of the article, and approved the final draft.

Field Study Permissions

The following information was supplied relating to field study approvals (i.e., approving body and any reference numbers):

Permits were provided by the National System of Conservation Areas of Costa Rica (SINAC-ACC-PI-LC-052-2 023).

Data Availability

The following information was supplied regarding data availability:

The data is available in the Supplemental File.

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
