# Peer review of "Host specificity of gastrointestinal parasites in free-ranging sloths from Costa Rica"

_PeerJ, doi:10.7717/peerj.19408_

## Round 0.1 · original submission · Major Revisions

Thank you very much for your manuscript titled “Host specificity of gastrointestinal parasites in free-ranging sloths from Costa Rica” that you sent to PeerJ.

This study presents very valuable and relevant information on the ecology of the sloth and its parasites.

As you will see below, comments from referee 1 suggest a major revision while reviewer 2 suggests a minor revision before your paper can be published. Given this, I would like to see a major revision dealing with the comments. Their comments should provide a clear idea for you to review, hopefully improving the clarity and rigor of the presentation of your work.

Reviewer 1 suggests clarifying some statistical and sample size issues as well as expanding the literature review, improving the figures and some points of discussion regarding the possible role of arthropods or environmental factors in parasite transmission.

Reviewer 2 has some comments on “parasite” terminology, sample collection, and interpretation of results when comparing urban vs. forest samples, and brings into the discussion the potential benefit of molecular identification of parasites.
Please note that we consider these revisions to be important and your revised manuscript will likely need to be revised again.

·

Basic reporting

Basic Reporting

Language and Clarity:
The manuscript is well written in clear and professional English. The text is concise, unambiguous, and maintains a scientific tone throughout. The authors have effectively communicated their findings, ensuring clarity and readability.

Literature References and Context:
The introduction provides a solid background on the ecological significance of sloths and the importance of studying gastrointestinal (GI) parasites in these species. Relevant prior studies are cited appropriately, though some references are outdated. The inclusion of more recent reviews on GI parasitology in wild mammals, particularly in sloths, would strengthen the background.

Article Structure, Figures, and Tables:
The manuscript follows the standard scientific format, including Introduction, Materials and Methods, Results, and Discussion. The figures and tables are relevant, well-labeled, and of sufficient resolution. However, Figure 4 (Parasite Richness Barchart) could benefit from additional statistical annotations (e.g., p-values or confidence intervals) to better convey the significance of the findings.

Raw Data Availability:
The authors provide supplementary materials, including raw data. However, a more detailed description of the dataset and its variables would improve reproducibility and transparency.

Self-Containment and Hypothesis Testing:
The study is self-contained and directly addresses its research questions. The findings align well with the stated hypotheses, but the discussion would benefit from more explicit linkage between results and broader ecological implications.

Experimental design

Scope and Relevance:
The research falls within the journal’s scope, addressing host specificity in GI parasites of free-ranging sloths. The study fills a knowledge gap concerning the parasite composition in sloths across different habitats.

Research Question and Knowledge Gap:
The study aims to determine whether parasite assemblages in sloths are primarily structured by host identity, habitat, or both. This is a meaningful and well-defined question that contributes to parasite ecology and wildlife health.

Technical and Ethical Standards:
The study adheres to ethical standards, with appropriate field study permits obtained from SINAC (Costa Rica). However, it is necessary to provide a document from an ethics committee on animal experimentation. The ethics committee approval code must be included in the text, and the document must be submitted as a supplementary file. Without this document, the article cannot be published. The sampling methods are rigorous, but more details on sample storage and handling procedures (e.g., temperature control, preservation duration) would enhance reproducibility.

Methods and Replicability:
The parasitological techniques used (e.g., Telemann and Sheather methods) are standard in the field. However, the description of statistical analyses, particularly the PERMANOVA test, should include assumptions checked (e.g., homogeneity of multivariate dispersions). The sample sizes per group (e.g., n=5 for some habitat categories) may be too small for robust statistical comparisons, requiring clarification on statistical power considerations.

Validity of the findings

Impact and Novelty:
While novelty is not a primary assessment criterion, the study significantly advances the understanding of sloth-parasite interactions. The discovery of potentially undescribed parasite species is particularly noteworthy.

Data Robustness and Statistical Soundness:
The dataset is generally robust, but statistical analyses could be strengthened by reporting effect sizes and confidence intervals. The lack of significant habitat effects may be due to limited sample size rather than true biological patterns.

Conclusions and Supporting Evidence:
The conclusions are well supported by the data and appropriately linked to the original research questions. The discussion effectively integrates findings with broader ecological concepts, but the potential role of intermediate hosts in parasite transmission could be explored further.

Additional comments

Recommendations for Improvement

Expand Literature Review: Include more recent studies on GI parasites in wild mammals, particularly in sloths, to provide a broader context.

Enhance Figure Annotations: Include statistical significance markers and confidence intervals in Figure 4.

Clarify Data and Statistical Analyses: Provide details on dataset variables, assumptions checked for PERMANOVA, and statistical power considerations.

Increase Sample Size Discussion: Address potential limitations due to small sample sizes in some habitat categories.

Further Investigate Intermediate Hosts: Expand discussion on the possible role of arthropods or environmental factors in parasite transmission.

Include Ethics Committee Approval: Add the ethics committee approval code to the text and submit the corresponding document as a supplementary file.

Final Assessment:
The manuscript is a well-executed study that provides valuable insights into sloth parasitology. With minor revisions to strengthen the statistical analysis, data presentation, and literature review, the paper would be suitable for publication.

Reviewer 2 ·

Basic reporting

This was a well-written study on the coprological (feces-based) analysis of sloth parasites in both wild forest and urban habitats. The conclusions based on the results presented are well-justified, and it was a pleasure and important to read. The results were well-contextualized in relation to the existing literature. and the figures and tables were of high quality. The authors found that the diversity of “parasites” (i.e., protozoa/nematodes/cestodes) was generally higher in ‘more wild’ sloths in primary forests than in those in urban habitats, and observed a higher diversity of novel parasites than has been documented in the literature to date—this is exciting to hear.

Experimental design

The experimental study was well-designed and executed, and the research questions posed were well-positioned to address our lack of understanding of GI protozoa/nematodes/cestodes associated with sloths. The research methods were described in detail, and with some exceptions (regarding clarifications, see below), provide enough information to replicate such a study.

Validity of the findings

The interpretation of the results were clear and logical, and proper statistical tests were performed to demonstrate significance and the strength of conclusions. The discussion and conclusions are grounded in the findings of the study, and considered the existing literature relevant to the topic and questions posed.

Additional comments

1. Throughout the manuscript, I would prefer that the authors be a bit more conservative in their use of the term “parasites” to describe the arthropods/nematodes/cestodes found as GI remnants of sloth feces. While the taxa identified are very commonly parasites to humans and other animals, in the context of sloth health, it’s not clear to me given my understanding of the limited literature that these arthropods/nematods/cestodes have any demonstrated fitness disadvantage for the sloth whether associated with in their GI tracts or not. Perhaps these organisms are just commensals. (It is interesting that ‘more wild’ forested sloths have a greater diversity of these critters—but not sure whether this is good/bad/neutral.) Unless fitness impacts have been measured, I think one should be careful as use of “parasite” it could incorrectly and inadvertently propagate the wrong notion that these organisms have negative consequences on sloths when we just have no clear idea. Using adjectives like “presumptive” or “putative” before “sloth parasites” could be a solution, as is possibly referring them to as “common parasites of animal life” just so the reader knows we are not referring to “parasites of sloths” per se.

2. Some clarifications are needed: for the fecal samples obtained from wild sloths, how did the authors confirm which species they were from (lines 107-109), and how long have those samples been deposited before sampling? Is it possible that the parasites observed migrated to them after defecation? (cf. arthropods being intermediate hosts (lines 233-241).

3. For the veterinary samples, were these sloths primarily from urban habitats (lines 109-110)? It was not clear, but I may have missed this. Also, could any of the observations made/results be a consequence of a difference in diet/eating habits between urban and forest sloths? Is there any observed difference in eating habits of primary forest sloths vs. urban sloths? A discussion of this as a factor in parasite diversity/load would be beneficial, perhaps in a paragraph before the coproparasitology paragraph starting on line 242.

4. Please clarify, e.g., in lines 156-160 and Figure 1: of the 22 B. variegatus samples, 12 were from primary forest and 10 were from an urban habitat – is primary forest = La Selva Biological Station and urban habitat = gray area in Puerto Viejo de Talamanca? And of the 16 C. hoffmanni samples, 5 were from undisturbed sites and 11 were from disturbed sites: what do you mean “undisturbed” vs. “disturbed” — is undisturbed in the green forested areas of Puerto Viejo de Talamanca and the disturbed in the gray areas of Puerto Viejo de Talamanca? In general, please annotate the map of Figure 1 with where you got fecal samples and for what sloth type. These annotations could also be harmonized with the “disturned” and “undisturbed” labels of Figure 4. This would go a long way in helping the reader understanding your results/study.

5. While I think a confirmation of fecal parasites by molecular methods is not needed/out of scope for this study, it would be beneficial if authors could describe if this would be (would have been) possible in their discussion, e.g., after lines 269-271. If this a “shortcoming” of the present study, please mention head-on—this should not preclude publishing these results as is in my opinion!

6. In lines 239-241: there is a typo or perhaps missing words between the combined words “plausiblearthropods” rendering it hard to understand the sentence: “…as both direct (e.g., arthropod carriers) and indirect (e.g., food-borne) routes of fecal-oral transmission are plausiblearthropods to serve as intermediate hosts of these GI parasites could help elucidate transmission pathways.”

7. Was there any specific reasons that samples were collected during March-July (line 96)? Some explanation would be beneficial.

---

## Round 0.2 · Minor Revisions

After reviewing this revised version of your manuscript, I see that the main comments suggested by the reviewers have been included. However, there are still some details that need to be clarified before having a final version that can be published.

As the reviewer emphasizes, it is necessary to clarify the controversy surrounding the biological concept of "parasitism." Furthermore, it is also necessary to consider amending the text to reflect "seasonal variations."

Reviewer 2 ·

Basic reporting

The authors have addressed the core of my concerns, thank you! Nevertheless, I have two responses to the authors' comments (see <<< for my response) that could improve the manuscript if addressed.

(1)
Original Comment: R2.4. Throughout the manuscript, I would prefer that the authors be a bit more conservative in their use of the term “parasites” to describe the arthropods/nematodes/cestodes found as GI remnants of sloth feces. While the taxa identified are very commonly parasites to humans and other animals, in the context of sloth health, it’s not clear to me given my understanding of the limited literature that these arthropods/nematods/cestodes have any demonstrated fitness disadvantage for the sloth whether associated with in their GI tracts or not. Perhaps these organisms are just commensals. (It is interesting that ‘more wild’ forested sloths have a greater diversity of these critters—but not sure whether this is good/bad/neutral.) Unless fitness impacts have been measured, I think one should be careful as use of “parasite” it could incorrectly and inadvertently propagate the wrong notion that these organisms have negative consequences on sloths when we just have no clear idea. Using adjectives like “presumptive” or “putative” before “sloth parasites” could be a solution, as is possibly referring them to as “common parasites of animal life” just so the reader knows we are not referring to “parasites of sloths” per se.

>>>Thank you for this thoughtful comment. You are absolutely right that we cannot attempt to quantify costs or benefit to the sloths as outcomes of interacting with these organisms based on our analysis or the available literature. We share your curiosity about whether the diversity of interactions should be interpreted as good, bad, or neutral. This comment prompted revisions that we made throughout the manuscript, beginning from the first lines of the Introduction and carrying through to our suggestions for future research.

>>>It is critically important to provide definitional clarity and consistency in our writing, especially using terms such as "parasite" that can connote different meanings depending on context. The Oxford Dictionary defines parasite as “an organisms that lives on, in, or with an organism of another species, obtaining food, shelter, or other benefit.” This is equivalent to the common use of the term in ecology and evolutionary biology—a parasite is defined by its resource niche and not by its tendency to harm the host. Many harmful diseases are caused by parasites, but not all parasites cause disease, and in some cases parasites can directly or indirectly benefit the host (e.g., due to genetic cofactors, virulence, immunity, nutrition, co-infection, or other environmental determinants of the outcome between the host and its parasite). We therefore use the term “parasites” in a manner consistent with its fundamental definition to identify organisms that inhabit the host as its environment and resource base. From the opening of the Introduction, we write: “Parasites are defined as organisms that live in or on other organisms and get nutrients from their host, living all or part of lives at the host’s expense—whether or not their tendency to cause disease or otherwise harm their host is consistent or severe (Méthot & Alizon 2014; Rózsa & Garay 2023)” (Lines 45-47).

<<<Given that the authors define how “parasite” is used, I might reluctantly accept this revision (though it’s possible numerous others in the field will form their own less than favorable opinions for this choice). While the OED is the authority on the English language, the use of “parasite” in this case is biological in context and not merely in the vernacular. Terminology is important. Like it or not, the way “parasite” is used biology is not decoupled from implying negative fitness effects to a host in the way that we could used the term in “everday” English. But I have voiced my concerns. I would ask that the authors make it clear and add, however, that their definition is essentially equivalent to “commensal”; while this might befuddle poor students who read this work and continue to propagate common misnomers in the literature, at least the term is defined as used.

>>>Then, in the Discussion, we write: “Second, evaluations of how GI parasites affect sloth health would require veterinary assessment. The parasites that we identified likely differ in the level and consistency harm they cause to their hosts—if any—which could vary as a function of both parasite diversity and load. Indeed, the mutualism-parasitism continuum holds that host-parasite interactions can vary from detrimental to commensal or beneficial depending on environmental context (Bronstein, 1994)” (Lines 287-292).

<<<This is good as this points to a standard reference on established biological terminology. Note however, consistent with my argument above, even this reference (and explicitly in Box 2), parasite is explicitly referred to as having a negative impact on their host(s) and not consistent with the author’s looser definition of parasite as quasi-inclusionary of commensals.


(2)
Original Comment: R2.10. Was there any specific reasons that samples were collected during March-July (line 96)? Some explanation would be beneficial.

>>>Thanks for prompting this clarification. This work was supported by a postdoctoral research fellowship from the Organization for Tropical Studies to the lead author, which took place during this time period. The funder is credited in our Acknowledgements, but what matters for the purpose of interpretation is that we state clearly that our research represents a snapshot comparison of host species across sites. Seasonal variation is an additional factor to consider in future research, which we addressed in the Discussion, “the transmissibility of each parasite can be modulated by both external environmental factors (e.g., contact between hosts in continuous vs. fragmented habitats) or internal variability in host susceptibility to infection (e.g., disparate levels of nutrition or environmental stress; Pool et al., 2016; Silva et al., 2013; Werner & Nunn, 2020)” (Lines 226-232).

<<<*Seasonal variation* was not explicitly stated, please modify your text as follows:
“the transmissibility of each parasite can be modulated by both external environmental factors (e.g., contact between hosts in continuous vs. fragmented habitats, season variations)…” (lines 230-231)

Experimental design

No comment

Validity of the findings

No comment

---

## Round 0.3 · accepted · Accept

After reviewing this revised version of your manuscript, I note that the two comments suggested by the reviewer have been included, especially regarding the concept of "parasitism" in the introduction. Therefore, I am satisfied with the current version and consider it ready for publication.